# Short-Chain Fatty Acids Modulate Permeability, Motility and Gene Expression in the Porcine Fetal Jejunum Ex Vivo

**DOI:** 10.3390/nu14122524

**Published:** 2022-06-17

**Authors:** Barbara U. Metzler-Zebeli, Simone Koger, Suchitra Sharma, Arife Sener-Aydemir, Ursula Ruczizka, Heinrich Kreutzmann, Andrea Ladinig

**Affiliations:** 1Unit of Nutritional Physiology, Department of Biomedical Sciences, University of Veterinary Medicine Vienna, 1210 Vienna, Austria; 2Christian-Doppler Laboratory for Innovative Gut Health Concepts of Livestock, University of Veterinary Medicine Vienna, 1210 Vienna, Austria; simone.koger@vetmeduni.ac.at (S.K.); suchitra.sharma@vetmeduni.ac.at (S.S.); arife.sener@vetmeduni.ac.at (A.S.-A.); 3Institute of Animal Nutrition and Functional Plant Compounds, Department of Farm Animal and Veterinary Public Health, University of Veterinary Medicine Vienna, 1210 Vienna, Austria; 4University Clinic for Swine, Department of Farm Animal and Veterinary Public Health, University of Veterinary Medicine Vienna, 1210 Vienna, Austria; heinrich.kreutzmann@vetmeduni.ac.at (U.R.); andrea.ladinig@vetmeduni.ac.at (H.K.); ruczizka@nutztier.at (A.L.); 5Association for Sustainable Animal Husbandry in Austria (NTÖ)–Project Animal Health Austria, 1200 Vienna, Austria

**Keywords:** fetus, microbial metabolites, mucosal sensing, small intestine, epithelial permeability, gut motility, late gestation

## Abstract

Postnatally, short-chain fatty acids (SCFA) are important energetic and signaling agents, being involved in host nutrition, gut imprinting and immune and barrier function. Whether SCFA exert similar effects during the late fetal phase has been insufficiently elucidated. This study aimed to evaluate whether the fetal jejunum senses SCFA and whether SCFA modify the muscle tension and epithelial permeability and related signaling in jejunal tissue from the porcine fetus in late gestation. Exposure of fetal jejunal tissue to a mix of SCFA (70 µmol/mL) in an organ bath for 20 min lowered the muscle tension. Moreover, SCFA decreased the transepithelial conductance while increasing the short-circuit current in the Ussing chamber, indicating reduced permeability and increased SCFA absorption. Gene expression in the tissues harvested from the Ussing chamber after 30 min indicated downregulation of the expression of receptors (i.e., *FFAR2* and *TLR2*), *MCT1* and tight-junction and adherens proteins, which may be a negative feedback response to the applied high SCFA concentration compared with the micromolar concentration detected in fetal gastric fluid. Taken together, our data demonstrate that the fetal jejunum senses SCFA, which trigger electrophysiological, muscle contraction and related gene transcription responses. Hence, SCFA may play a role in prenatal gut nutrition and imprinting.

## 1. Introduction

Fetal nutrition almost exclusively occurs via the umbilical cord, providing ready-to-use nutrients for the developing organism [1,2]. Towards the end of the pregnancy, the fetus starts drinking amniotic fluid [2,3], which serves as a reservoir of fluid and nutrients for the fetus, containing proteins, carbohydrates, electrolytes, immunoglobulins and vitamins from the mother [4]. Simultaneously, the inherent antibacterial properties in amniotic fluid help to protect the fetus from infectious agents [4]. Contrasting the latter, the presence of bacterial DNA in amniotic fluid [4] allows for assuming that gut microbial activity and compounds from microbial metabolism may play a role in fetal development in late gestation. Likewise, meconium contains relevant concentrations of SCFA (especially acetate and propionate), which supports that microbial action in a piglet’s gut commences before birth [3,5,6]. Short-chain fatty acids (SCFA) may also reach the fetuses via the placenta, as small concentrations are present in maternal blood [7,8]. Postnatally, SCFA, such as acetate, propionate, butyrate, isobutyrate, valerate, isovalerate and caproate, are recognized as important energetic and signaling agents. They are involved in host nutrition, imprinting of the immune system, cell metabolism, barrier function and intestinal motility [9,10,11]. However, their mechanistic action on the host can differ depending on the type of SCFA and molar proportions in which the SCFA occur. In this line, SCFA-sensing free fatty acid transporters (FFAR) 2 and 3 and monocarboxylate transporters (i.e., sodium-dependent monocarboxylate transporters (SMCT) and monocarboxylate transporter (MCT)-1) have different affinities for the various straight- and branched-chain fatty acids [11], and the results from our group showed that FFARs and SCFA transporters are highly expressed in the porcine small intestine from day 7 of life [12]. Measurement of SCFA in the cecal and abdominal blood of catheterized neonatal piglets support efficient paracellular and transcellular absorption [9]. However, whether SCFA can already exert nutritional, developmental and health benefits in utero has been investigated little so far. By assuming that SCFA contribute to prenatal imprinting, this would strengthen the concept of the “mother-offspring axis” and the importance of the maternal diet not only for major diet-derived nutrients but also for microbial metabolites. Preterms and small-for-their-age newborns are prone to gut inflammatory, microbial and metabolic disturbances [13]. Preterm infants may benefit from an oral administration of SCFA, as has been suggested as a therapeutic approach in the treatment of inflammatory bowel diseases [14]. Therefore, oral SCFA may support the missing intra-uterine maturation of the intestine if the mucosa responds appropriately at this stage of life. In utero, after swallowing amniotic fluid and after oral administration to the newborn, the nutrients come in contact with the gastric and small intestinal epithelium first and, if recognized, trigger receptor-mediated signaling chains or absorption. We recently showed in a porcine ex vivo model that the neonatal jejunum strongly responds to SCFA by increasing muscle contractibility [10], which justifies investigating the SCFA sensing in the fetal small intestine. In the unborn piglet, the gut matures substantially in the last days of gestation, showing similar deficiencies in the intestinal function and barrier to human preterms, thus representing a suitable model for human infants born prematurely [15].

This study aimed to evaluate whether the fetal jejunum senses SCFA and whether SCFA modify the muscle tension and epithelial permeability in the jejunal tissue from a porcine fetus at a functional level in late gestation using an ex vivo approach. We hypothesized that, similar to the neonatal piglet [10], SCFA trigger an upregulation of the muscle contractibility and improve ion uptake and epithelial barrier function in the fetal jejunum, which is mediated via alterations in the expression of mucosal FFARs, transcription factors and histone deacetylase.

## 2. Materials and Methods

### 2.1. Animals

In total, four fetuses from four sows which belonged to the control group of a different study investigating porcine reproductive and respiratory syndrome virus (PRRSV) infection in late gestation were used in the present experiment. All procedures involving animal handling and treatment were approved by the institutional ethics committee of the University of Veterinary Medicine Vienna and the national authority according to the Law for Animal Experiments, Tierversuchsgesetz in Austria (BMWFW-2021-0.117.108).

The sows were moved to the animal facility of the University Clinic for Swine on day 77 of gestation, group-penned (pen size 2.5 × 7 m) with straw as bedding material and fed a commercial gestation diet (Königshofer GmbH, Ebergassing, Austria; metabolizable energy: 13.0 MJ/kg; crude protein: 15.2%; ash: 3.5%; ether extracts: 4.0%; crude fiber: 4.5%, calcium: 1.0%; phosphorus: 0.6%; lysine: 1.0%; methionine: 0.33%; threonine: 0.65% and sodium: 0.3% on an as-fed basis), and hay and straw was available ad libitum. On gestation day 106 (±2), an intravenous injection of ketamine (Narketan^®^ 100 mg/mL, Vetoquinol Österreich GmbH, Vienna, Austria; 10 mg/kg body weight) and azaperone (Stresnil^®^ 40 mg/mL, Elanco GmbH, Cuxhaven, Germany; 1.5 mg/kg body weight) was applied before the gilts were euthanized by an intracardiac injection of T61 (embutramide, mebezonium iodide and tetracaine hydrochloride; Intervet GesmbH, Vienna, Austria; 1 mL/10 kg body weight). The gravid reproductive tract was removed intact, placed in a trough and rinsed with tap water in order to remove the maternal blood. Several minutes passed from removal to opening of the uterus to ensure the death of each fetus. All fetuses were clinically dead when sampled. Detailed necropsies were performed as described earlier [16]. Two male and two female fetuses were selected from the left uterine horn close to the ovary. Because the myometrium and fetal membranes are often already broken during necropsy, collection of sterile fluid from the amniotic sac can be challenging. Therefore, we collected the fluid contained in the stomach of the fetuses by puncturing the stomach with the use of a sterile needle and syringe. The fetus swallows amniotic fluid in late gestation; hence, it can be assumed that the liquid in the stomach primarily consisted of amniotic fluid. The abdomen of the fetus was opened, and the whole gastrointestinal tract was removed, placed into ice-cold modified Krebs–Henseleit buffer for transport to the laboratory (Figure 1 and Table 1).

The buffer was pre-gassed with carbogen gas (95% O_2_ and 5% CO_2_) to allow respiration of the tissues. The time elapsing from euthanization of the mother sow until the start of the equilibration phase in the Ussing chambers and tissue bath was not longer than 45 min. In the laboratory, the jejunum was identified, and two gut segmental samples 20 cm in length were taken at mid-jejunum: one cranial from the midst, which was used in the organ bath, and the other one distal to the midst towards the ileum, which was used in the Ussing chamber system. The tissues that were mounted into the Ussing chamber were collected at the end of the experiment for RNA isolation.

### 2.2. Short-Chain Fatty Acid Solution

Physiological concentrations of the predominant SCFA acetate, propionate, butyrate, valerate and caproate as well as the branched-chain SCFA isobutyrate and isovalerate (Sigma-Aldrich, St. Louis, MO, USA) were used to evaluate the SCFA effect at the jejunal tissue. The stock SCFA solution contained 1.96 mmol SCFA/mL, which was added to the organ bath and Ussing chambers to reach a final concentration of 70.5 µmol/mL in each chamber (Table 2) [10].

This concentration is more representative for concentrations found in the meconium and large intestine of neonatal piglets [10], surpassing the concentrations found in the fluid collected from the fetal stomach (Table 3). The high concentration was selected to be high enough to provoke a physiological response, similar to our previous study [10]. The molar ratios for the three main SCFA were 71.4% acetate, 10.9% propionate and 14.4% butyrate.

### 2.3. Measurement of Jejunal Motility

The jejunal segment of each fetus was cut into four tube pieces 1 cm in length, and their lumen was flushed with Krebs–Henseleit buffer (pH 7.4; Table 1), warmed to 37 °C and pre-gassed with carbogen gas [10,17]. Each of the four tube pieces was carefully tied at both ends with silk suture without closing the lumen. One end of the suture was attached to a hook that was submerged, together with the tube piece, in buffer in a 20-mL water-jacketed chamber which was maintained at 37 °C and continuously gassed with carbogen gas (95% oxygen and 5% carbon dioxide). The silk suture on the opposite end of the jejunal tube piece was attached to a force transducer (SEN-03-34, MDE, Heidelberg, Germany) which was connected to a four-channel bridge amplifier (EXP-SG-4, MDE, Heidelberg, Germany) to continuously collect the muscle tension data. The tension of each tube piece was adjusted to an initial tension of 1.0 g or 10 mN. The tube pieces were equilibrated for 20 min, and the buffer was replaced 3 times in 5-min intervals. A defined volume of 12 mL Krebs–Henseleit buffer was then provided in each chamber. Immediately thereafter, the viability of the jejunal and ileal tube pieces was tested by contraction with the addition of acetylcholine (ACh), reaching a concentration in each chamber of 10 µM. When the contraction to ACh was stable after 10 min, the SCFA solution was added, and the response was recorded as positive or negative changes in tension for 20 min from when the tension returned to basal values. The responses to the SCFA were calculated as the maximum decrease or increase in tension in comparison to the mean tension 1 min prior to the addition of the SCFA solution. The increase or decrease in tone was measured in grams, and the percent relaxation induced by the SCFA was calculated as the percent decrease or increase from the level of tone prior to the addition of the SCFA solution.

### 2.4. Measurement of Gut Electrophysiological Parameters

The effects of SCFA on the jejunal electrophysiology were tested in four replicates per fetus, using a similar procedure to that described by Baskara et al. [17]. In omitting the first centimeter, six consecutive jejunal tube pieces were prepared, which were opened at the mesenterium and rinsed with modified Krebs–Henseleit buffer (Table 1) to remove digesta particles [17]. The thinness of the jejunal tissue did not allow stripping off the serosa, and hence the jejunal wall remained intact. Jejunal tissue pieces of a little larger than the exposed surface area of 0.9 cm^2^ were cut, mounted in the Ussing chambers, and rested for an equilibration of 10 min under open-circuit conditions. The tissues were continuously gassed with carbogen gas and kept at 37 °C. The SCFA solution or modified Krebs–Henseleit buffer as controls were added to the mucosal side 5 min after clamping the tissues to a potential difference of zero. In addition, modified Krebs–Henseleit buffer was added as a volume adjustment to the serosal side. Gut electrophysiological measures, including the short-circuit current (I_SC_) and transepithelial tissue conductance (G_T_) indicators for the net ion flux and tissue permeability, were recorded for 30 min after the addition of the respective solution. The chemical effect on the mucosal nutrient flux and permeability was assessed by comparing the I_SC_ and G_T_ for 1 min before the addition of the SCFA and the peak current and resistance response of the exposed tissue (ΔI_SC_ and ΔG_T_) obtained within the 30 min after the addition of the SCFA or buffer as a control treatment. After the 30 min, the jejunal tissue was harvested from each chamber, snap-frozen in liquid nitrogen and stored at −80 °C until RNA isolation.

### 2.5. Analysis of SCFA

Gas chromatography (GC) was used to measure SCFA in the stock solution and in the amniotic fluid. The amounts of single SCFA in the stock solution were determined essentially as described by Yosi et al. [18]. In the amniotic fluids (600 µL), single SCFA were measured after extraction by adding 200 μL of 1.8 mol hydrochloric acid and internal standard 4-methylvaleric acid (Sigma-Aldrich, Vienna, Austria). The samples were vortexed and centrifuged at 20,000× *g* for 20 min at 4 °C. The clear supernatant was transferred into glass vials for the GC.

### 2.6. Measurement of the Gene Expression

In total, 22 target genes in the jejunal tissue samples coding for SCFA-sensing receptors, transporters, transcription factors and cytokines were analyzed (Appendix A). Previously published primers for amplification of the target and housekeeping genes were checked for accuracy [19,20,21], or the primer sets were newly designed. Both tasks were accomplished using PrimerBLAST (www.ncbi.nlm.nih.gov/tools/primer-blast/ (accessed on 4 May 2022)). The jejunal tissue samples were pulverized in liquid nitrogen using a mortar and pestle. The total RNA was isolated from approximately 20 mg using mechanical homogenization and the RNeasy Mini Kit (RNeasy Mini Qiacube Kit, Qiagen, Hilden, Germany) as well as treatment with DNase I (Invitrogen^TM^ TURBO DNA-free^TM^ Kit; Thermo Fisher Scientific Inc., Waltham, MA, USA). The RNA was evaluated with a Qubit fluorometer using a Qubit RNA HS Assay for quantification and a Qubit RNA IQ Assay (Qubit 4 Fluorometer; Qubit RNA HS Assay Kit; Qubit RNA IQ Assay Kit, Thermo Fisher Scientific Inc., Waltham, MA, USA) for quality control. If the RNA integrity number was below eight, the RNA was newly isolated from the respective sample. Complementary DNA (cDNA) was synthesized using a high-capacity cDNA RT kit following the manufacturer’s protocol (High-Capacity cDNA Reverse Transcription Kits; Thermo Fisher Scientific Inc., Waltham, MA, USA). Amplification and quantification of the cDNA was performed with innuMIX qPCR DSGreen Standard (IST Innuscreen GmbH, Berlin, Germany) on a qTower^3^ 84 system (Analytik Jena GmbH, Jena, Germany). The pipetting for qPCR was conducted using a robot (epMOTION 5075^TMX^, Eppendorf AG, Hamburg, Germany). Each reaction including a negative template control and RT minus controls contained a master mix (7 µL) including 2.5 µL of innuMIX qPCR DSGreen Standard Mastermix, forward and reverse primers (200 nM) and 25 ng of a DNA template. After an initial denaturation step at 95 °C for 2 min, 40 cycles of 95 °C for 30 s followed, as well as primer annealing and elongation at 60 °C for 60 s. Melting curve analysis was performed to verify the specificity of the PCR amplification. For the calculation of the relative gene expression, *ACTB*, *GAPDH*, and *B2M* were identified to be the most stably expressed HKGs after assessment with NormFinder and BestKeeper and were used as endogenous controls [19]. The relative gene expression was calculated using the 2^−ΔΔCq^ method, for which the geometric mean of the HKG was used for normalization of the Cq values to determine the ΔCq values. The sample with the highest expression and hence lowest ΔCq value of the respective targeted gene was used for the second normalization in order to calculate the ΔΔCq and respective 2^−ΔΔCq^. For the absolute gene expression, standard curves were generated using serial dilutions (from 10^−3^ to 10^−7^ molecules/µL) of the purified products (QIAquick PCR Purification Kit, Qiagen, Hilde, Germany) and quantified PCR products (Qubit™ dsDNA HS Assay Kit, Thermo Fisher Scientific, Waltham, MA, USA) generated by qPCR from the samples to assess the amplification efficiency (Appendix A).

### 2.7. Statistical Analysis

The Shapiro–Wilk test and UNIVARIATE procedure in SAS (Version 9.4; SAS Stat Inc., Cary, NC, USA) were used to test the residuals for the data from the organ bath, Ussing chambers and gene expression experiments for normal distribution. For the jejunal motility parameters, the data were analyzed as repeated measures over time to study whether the addition of the SCFA modified the basal tension of the jejunal tissue pieces using the MIXED procedure in SAS. A second model was used to analyze the gut electrophysiological and gene expression data. The second model included the fixed effect of the SCFA addition and the random effects of the chamber and sex. In the repeated and random models, the experimental unit was “chamber replicate within fetus”. The degrees of freedom were approximated by the method of Kenward–Roger (ddfm = kr). The least squares means were computed, and significance was declared at *p* ≤ 0.05 with trends at 0.05 < *p* ≤ 0.10. Pairwise comparisons among the least squares means were performed using the probability of difference option in SAS. The MEANS procedure was used to determine the mean values of the SCFA in the amniotic fluid, proportional change in contractibility between the basal muscle tension and tension after SCFA addition as well as for absolute gene expression in the jejunal tissue.

## 3. Results

The fluid that was extracted from the fetal stomachs contained small amounts of total SCFA, acetate and propionate in the micromolar range, amounting to 0.35, 0.34 and 0.01 µmol/mL, respectively (Table 3). The addition of SCFA reduced the muscle tension of the jejunal tube pieces compared with the basal muscle tension, which corresponded to a relaxation effect of 30.1 ± 3.87% (Figure 2; *p* < 0.001).

The addition of the SCFA also modified the electrophysiological data. The SCFA increased the I_SC_ by 126.8% within the 30 min after their addition compared with the 1% change in the I_SC_ of the control (Table 4; *p* < 0.001). Likewise, the SCFA addition reduced the G_T_ by 40.8% within the 30 min after their addition, whereas the G_T_ decreased by only 5.1% in the control chambers within the 30 min of the test period (*p* < 0.001).

The jejunal tissue pieces from the Ussing chambers were used in the gene expression experiment to assess whether the differences in jejunal functioning observed after the SCFA addition were also detectable at the gene expression level after 30 min of incubation. The absolute expression of the target genes is presented in Table 5.

In the incubated jejunal tissue, the SCFA addition decreased the expression of *FFAR2*, *TLR2* (*p* < 0.05) and *EGFR* (as a trend, *p* < 0.10) by 16.8%, 27.3% and 10.9%, respectively, compared with their expression in the control tissue pieces (Figure 3).

From the transporters, SCFA tended (*p* < 0.054) to decrease the expression of *MCT1* by 12.6% compared with the control (Figure 3). Moreover, the SCFA downregulated the expression of the tight junction proteins *OCLN* (as a trend, *p* = 0.055), *CLDN4*, *CDH1* and *JAML* (*p* < 0.05) by 19.6%, 42.9%, 16.7% and 39.4%, respectively, compared with the buffer addition as control (Figure 4). With regard to the inflammatory signaling, the SCFA addition reduced the expression of *IL18* compared with the control by 19.7% (*p* = 0.016; (Figure 4)).

## 4. Discussion

The effects of the SCFA were immediately traceable after addition in real time in the Ussing chamber and organ bath experiments. Hence, the present results confirmed our hypothesis that the fetal jejunum is capable of sensing SCFA in late gestation. The results indicate an enhanced barrier function and, indirectly as a sodium co-transporter, increased SCFA absorption. By contrast, the SCFA did not increase muscle contractibility, as we assumed from our findings for the neonatal jejunum, which was incubated with the same SCFA solution in a previous study [10]. Instead, the SCFA caused muscle relaxation in the fetal jejunal tissue. Despite the short incubation time of 30 min, the SCFA-induced changes were not only evident at the functional protein level, but they were also noticeable at the gene expression level, involving the downregulation of certain SCFA receptors, monocarboxylate transporter-1, cytokine and tight-junction proteins. The gene expression results allowed characterizing potential underlying signaling routes for the altered epithelial permeability. Note that the SCFA concentration in the Ussing and organ bath chambers was in the millimolar range and largely surpassed the micromolar concentrations measured in the fluid collected from the stomachs of the fetuses. This may have led to a certain overload of the jejunal tissue and may have triggered negative feedback mechanisms. As a next step, it would therefore be reasonable to investigate the dose–response relationship using SCFA concentrations from as low as we found in the fluid extracted from the fetal stomachs up to the dosage from the present study. In terms of the SCFA in the fetal gastric fluid, it is important to be aware that, as lipid-soluble molecules, a small concentration of SCFA may have already been absorbed by the stomach epithelium. The present electrophysiological data for the fetal jejunum may support that SCFA absorption by the stomach epithelium may be possible. Nevertheless, the collection of fluid from the fetal stomach was our method of choice to collect amniotic fluid in order to avoid contamination by external fluids from the opened body cavity of the sow. Overall, as this study has demonstrated, microbial metabolites that are present in the lumen of the fetal small intestine may play a role in the prenatal priming and maturation of the fetal gut. It is worth emphasizing again that we observed these effects in the small intestine and not in the fetal hindgut, where microbial–host interactions may be more conceivable as the lumen was already filled with dark brownish digesta at the time of sampling (Figure 1).

Part of the SCFA detected in the gastric fluid in the present study can be assumed to have originated from the mother dam’s blood, with the other part originating from microbial metabolism in the amniotic fluid. The SCFA measured in the amniotic fluid were mainly acetate and propionate in the present study, although in one sample, caproate at 0.01 µmol/mL was detected. Hence, the fetal porcine gut was exposed to acetate and propionate (and caproate) in utero, which may have activated the sensing capacities for these SCFA in late gestation. Since the SCFA originating from the mother’s blood can be assumed to be more diverse, we applied the full panel of SCFA, which stimulates the natural condition when SCFA are produced during carbohydrate or protein fermentation and diffuse into the blood after absorption and hepatic metabolism. Note that pigs differ from humans in their type of placentation, having an epitheliochorial and diffuse type of placenta compared with the discoid, hemochorial human placenta, which influences placental permeability and nutrient transfer from the mothers to fetuses [22]. However, potential differences in transfer are irrelevant for the present fetal model, as we added the SCFA artificially in the Ussing and organ bath chambers.

The fetal jejunum responded differently to exposure to the SCFA solution compared with the neonatal jejunum [10], suggesting an inhibition of muscle contractibility. Different reasons may underlie this discrepancy. These may have been related to the state of immaturity of the fetal jejunum in late gestation, the inferior role of absorptive processes in the jejunum for fetal nutrition or negative feedback to the high SCFA concentration. Opposite to the latter assumption, the decreased muscle contractibility could also indicate that the passage of the jejunal content was slowed down to increase the time for absorption of the SCFA. For interpretation of the results, it should be kept in mind that we worked with separate, consecutively cut 1-cm long tube pieces. As the SCFA response was provoked in all replicates, the present observation provided a first idea of whether SCFA are recognized by and provoke effects at the mucosa of the fetal jejunum. With regard to factors that may influence the mucosal response to SCFA, aside from the state of maturity, it may be conceivable that growth factors in colostrum and mature milk or factors from the developing microbiota may alter the response of the jejunum to SCFA postnatally compared with the late fetal phase.

In contrast to the muscle contractibility, the gut electrophysiological measures were in line with the postnatal situation, indicating increased positive ion flux and potentially increased SCFA absorption as well as reduced epithelial permeability [10]. SCFA are absorbed transcellularly via carrier proteins and paracellularly via pores formed by tight-junction proteins [23]. Postnatally, the sodium-coupled transporters (SMCT) are the major SCFA transporters in the porcine jejunum, whereas monocarboxylate transporter (MCT)-1 dominates in the large intestine [24]. According to the present results for absolute gene expression, MCT-1 seems to play an equally important role in SCFA absorption in the fetal jejunum to SMCT-1. Either way, the less negative I_SC_ pointed towards an increased co-transport of sodium (Na^+^) ions due to the activation of the sodium-coupled monocarboxylate transporters (SMCT) 1 and 2, where both are expressed at the apical membrane of the small intestine [24,25]. The real-time data from the Ussing chamber experiment indicated that the I_SC_ continuously increased until plateauing at 20 min of exposure, suggesting a continuous absorption of sodium ions. SMCT-1 and -2 differ in their stoichiometry for sodium ion and SCFA transport. SMCT-1-mediated transport is electrogenic (Na^+^:SCFA stoichiometry = 2:1), whereas SMCT-2-mediated transport is electroneutral (Na^+^:SCFA stoichiometry = 1:1) [25,26]. As SMCT-1 showed greater expression than SMCT-2, it may be assumed that SMCT-1 proportionally contributed more to the ΔI_SC_ of 37.5 µA/cm^2^ than SMCT-2. In considering that the I_SC_ represents the net ion flux across the epithelium [27], another scenario may have contributed to the observed change in I_SC_. Aside from increased Na^+^ absorption, there may have been decreased anion secretion (i.e., chloride and bicarbonate ions), which would fit the continuing decrease in G_T_ which accompanied the increase in I_SC_. The results for the expression of the MCTs did not help explain our gut electrophysiological findings. All three SCFA transporters were highly expressed at the fetal jejunal mucosa. In contrast to the increase in I_SC_, *SMCT1* and *SMCT2* were similarly expressed at the fetal mucosa in both treatment groups. It can be speculated whether the SCFA response was mainly at the functional protein level and did not lead to transcriptional changes after 30 min of exposure. It may also be probable that at the maturational stage of the fetus, the transporter expression may not respond to changes in ligand availability. Third, as we added a high SCFA concentration, the jejunal tissue may have tried to control the SCFA uptake by not increasing the expression of the SMCTs. The latter assumption may be supported by the trend for a decrease in *MCT1* expression in the jejunal tissue when exposed to SCFA, potentially indicating negative feedback to limit SCFA absorption. We did not measure the SCFA concentration in the serosal buffer, which may have provided a certain idea about the dimension of SCFA absorption. While not being directly transferable, the findings in the laying hens from our group using a comparable ex vivo approach and SCFA concentration showed similar trends for I_SC_ and G_T_, which was accompanied by higher SCFA concentrations in the serosal buffer in the SCFA groups compared with the control group [18].

The SCFA appeared to improve the mucosal barrier function, as indicated by the SCFA-induced decrease in G_T_. This finding may be beneficial for newborns with regard to the prevention of antigens crossing the mucosal barrier. Paracellular permeability is regulated by pores that are formed by tight-junction proteins, including transmembrane proteins, such as occludin (OCLN), claudins (CLDN) and junctional adhesion molecule (JAM), peripheral membrane adaptor proteins (e.g., zonula occludens-1 (ZO)), as well as adherens junction proteins (e.g., E-cadherin (CDH)) [23]. The present results for G_T_ may suggest that the tight-junction proteins formed a physical barrier, reducing the mucosal to serosal flux of extracellular components. This assumption would then lead to the assumption that SCFA may have been absorbed to a greater degree via the transcellular route.

The downregulation of the expression of several of these components (i.e., expression of *OCLN* (trend), *CDH*, *CLDN4* and *JAML*) in the fetal jejunum after exposure to SCFA for 30 min shows that the inhibiting effect of the SCFA on paracellular permeability was not only at the functional level but was already transmitted during the transcription of genes. Improved barrier function may be desirable in newborns from a gut health perspective, especially in the early born or neonates born too small for their age. However, it needs to be evaluated whether this decreased jejunal permeability compromises the uptake of essential nutrients and immunoglobulins before a recommendation can be given as to whether external SCFA may provide an extra benefit for barrier function in newborns.

The SCFA may initialize the alterations in muscle tension, ion flux and permeability via multiple pathways and receptors, including the activation of muscarinic ACh receptors, calcium release via L-type calcium channels, or via activation of histamine, nicotine and serotonin receptors [28,29,30,31]. In the present study, we investigated several receptors (i.e., FFAR-2, FFAR-3, HCAR-2 epidermal growth factor receptor (EGFR) and Toll-like receptor (TLR)-2), as well as the SCFA-induced inhibition of histone deacetylase (HDAC), which have been shown to respond to SCFA at micro-to-millimolar ranges [32,33,34]. These receptors differ in their affinities for the single SCFA. Therefore, it can be assumed that all SCFA probably signaled in the present experiment. Applying single SCFA will likely create different receptor responses as in the present study, where the SCFA compete for receptor binding positions. Of note, instead of stimulating receptor expression, the present SCFA mix and concentration mainly signaled via downregulation of the expression of *FFAR2*, *TLR2* and *EGFR* (as a trend) after 30 min of exposure. It is probable that the high SCFA concentration triggered the downregulation as a negative feedback response to the high stimulation of the active receptors, which may have led to the decrease in *MCT1* and tight-junction and adherens junction protein expression. With regard to the tight-junction and adherens junction proteins, their expression is closely related to activation of the pattern recognition receptors, such as TLR, which often signals via the key proinflammatory transcription factor NF-κB. In the present study, the SCFA did not alter *NFKB* expression in the jejunal tissue. However, other receptors were triggered and downregulated, which may have modulated the transcription of the *NFKB* gene [33,35]. In this regard, FFARs signal via activation or phosphorylation of adenylate cyclase and phospholipase C and alterations in intracellular Ca^2+^ ions and cAMP levels, and they are involved in the regulation of proinflammatory signal transduction [33]. FFAR-2 has a high affinity for acetate, propionate, butyrate and especially for valerate [33], which were all present in the present SCFA mix. Sodium butyrate has been reported to induce the expression of defensins, such as pBD3 and pEP2C, via TLR-2 and EGFR in IPEC J2 cells [32]. In the present study, we observed a decrease in receptor expression due to the SCFA but not in the two defensins. It can be speculated whether the SCFA signal was already transmitted or whether the fetal tissue was too immature to induce a similar reaction. Nevertheless, the SCFA downregulated the expression of proinflammatory *IL18*, which has been previously linked to the expression levels of pBD3 [32]. Since the expression levels of the two measured proinflammatory factors *NFKB* and *IL18* were not upregulated, and that of *IL18* was even downregulated, it may be assumed that the largely higher concentration of SCFA in the applied mix in comparison with the SCFA concentration measured in the gastric fluid did not act as a proinflammatory stimulus at the fetal jejunal mucosa.

## 5. Conclusions

The results of the present ex vivo study demonstrate that the fetal jejunum in late gestation is capable of sensing SCFA and triggering physiological responses at the functional protein level and gene expression level. The SCFA addition reduced the mucosal permeability while potentially increasing SCFA absorption. Moreover, the SCFA led to muscle relaxation in the fetal tissue. These findings support that SCFA may play a role in prenatal function and imprinting in late gestation. The downregulation of the expression of *FFAR2*, *TLR2* and *EGFR* as well as *MCT1* and tight-junction and adherens junction proteins after 30 min of exposure may have been a negative feedback response to the high concentration of SCFA compared with the concentration at the micromolar range found in the fluid extracted from the fetal stomachs. As a next step, inhibitor studies should be conducted in the future to clarify the exact signaling that occurred at the jejunal mucosa.

## Figures and Tables

**Figure 1 nutrients-14-02524-f001:**
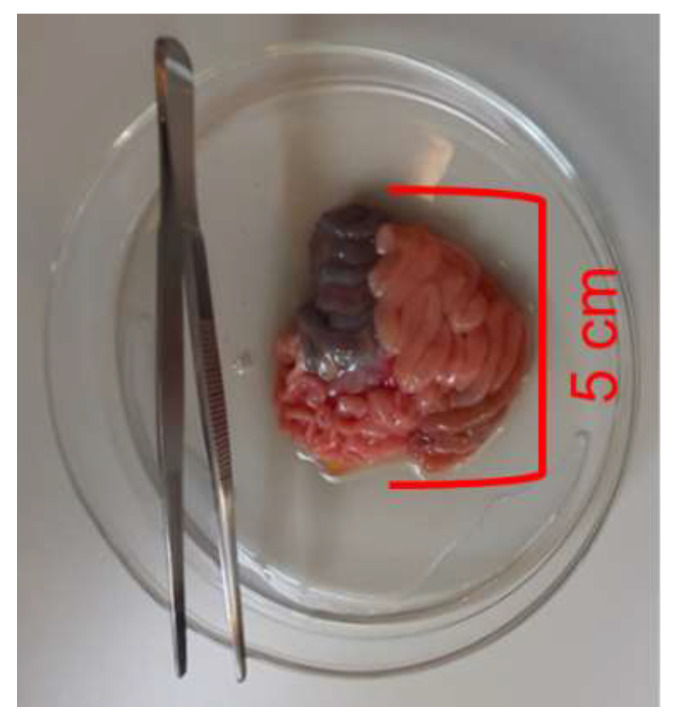
Fetal gut convolute.

**Figure 2 nutrients-14-02524-f002:**
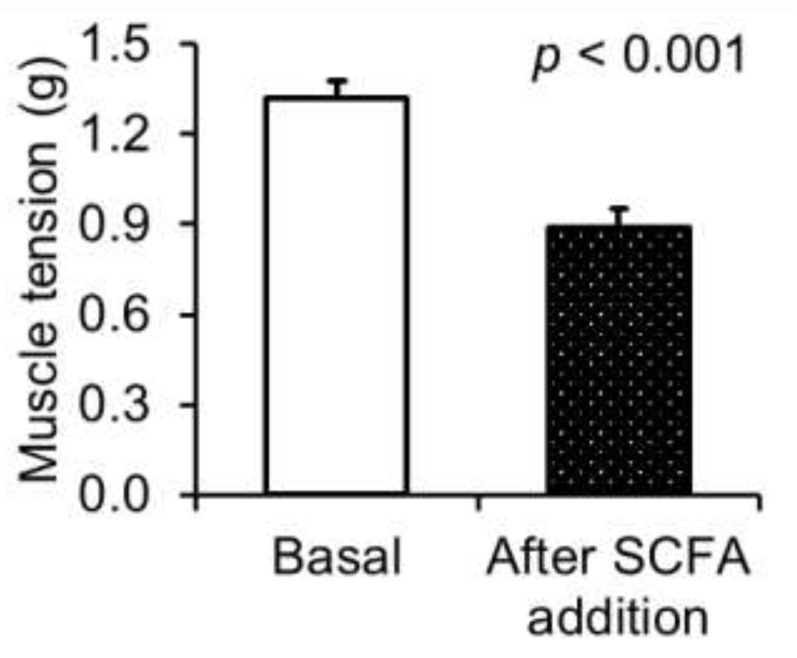
Effect of short-chain fatty acids (SCFA) on muscle tension of the fetal porcine jejunum. Jejunal muscle tension before (basal) and after the addition of the SCFA solution is shown. Values are least squares means and standard error of means (SE), where *n* = 16.

**Figure 3 nutrients-14-02524-f003:**
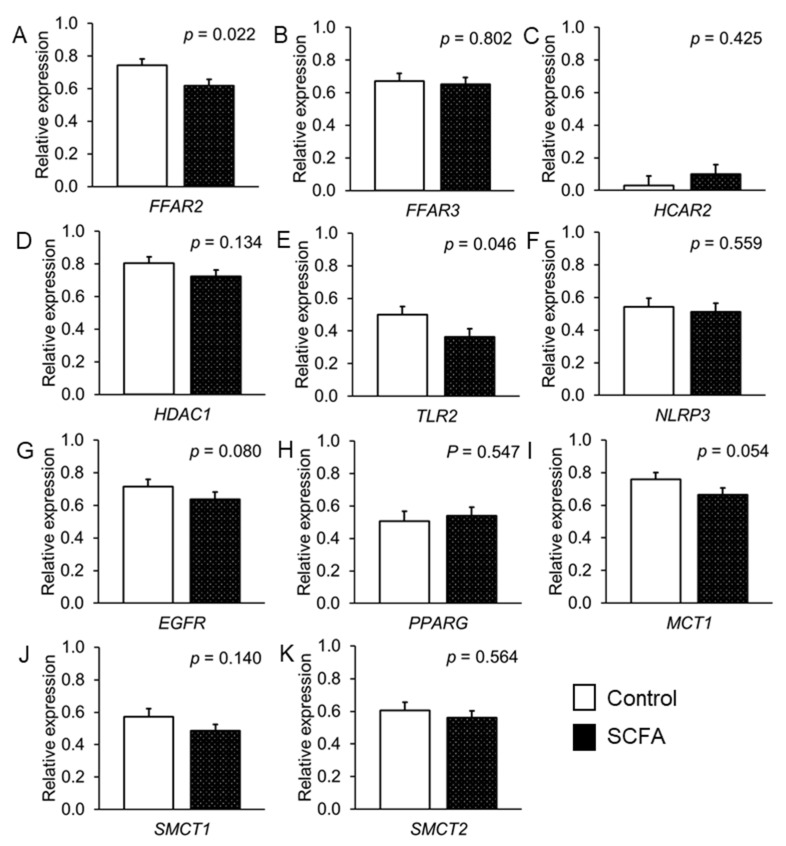
Effect of short-chain fatty acids (SCFA) on expression of SCFA receptors and transporters and the transcription factors in the fetal porcine jejunum. Values are least squares means and standard error of means (SE), where *n* = 16: (**A**) *FFAR2* = free fatty acid receptor-2; (**B**) *FFAR3* = free fatty acid receptor-3; (**C**) *HCAR2* = hydroxycarboxylic acid receptor 2; (**D**) *HDAC1* = histone deacetylase-1; (**E**) *TLR2* = Toll-like receptor-2; (**F**) *NLRP3* = NLR family pyrin domain containing 3; (**G**) *EGFR* = epidermal growth factor receptor; (**H**) *PPARG* = peroxisome proliferator activated receptor-gamma; (**I**) *MCT1* = monocarboyxylate transporter-1; (**J**) *SMCT1* = sodium-coupled monocarboxylate transporter-1; (**K**) *SMCT2* = sodium-coupled monocarboxylate transporter-2.

**Figure 4 nutrients-14-02524-f004:**
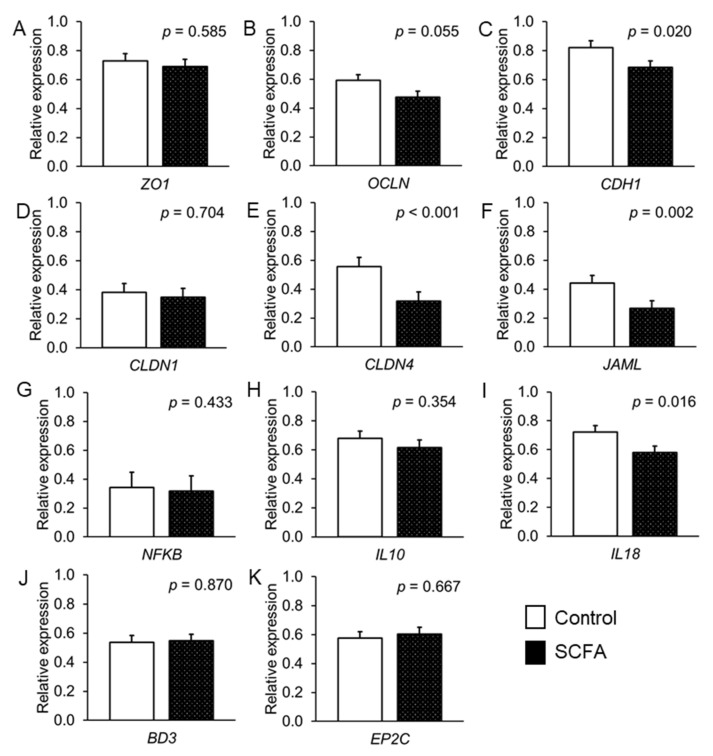
Effect of short-chain fatty acids (SCFA) on expression of tight-junction and adherens junction proteins, cytokines and defensins in the fetal porcine jejunum. Values are least squares means and standard error of means (SE), where *n* = 16: (**A**) *ZO1* = zonula occludens-1; (**B**) *OCLN* = occludin; (**C**) *CDH1* = E-cadherin; (**D**) *CLDN1* = claudin-1; (**E**) *CLDN4* = claudin-4; (**F**) *JAML* = junction adhesion molecule like protein; (**G**) *NFKB* = nuclear factor kappa B; (**H**) *IL10* = interleukin-10; (**I**) *IL18* = interleukin-18; (**J**) *BD3* = beta-defensin-3; (**K**) *EP2C* = epididymis protein 2 splicing variant C.

**Table 1 nutrients-14-02524-t001:** Composition of (modified) Krebs–Henseleit buffer used in organ bath and Ussing chambers ^1^.

Components	Concentration (g/L)	Molarity (mM)
Krebs–Henseleit buffer used in organ bath		
NaCl	6.90	118.1
NaHCO_3_	2.10	25.0
KCl	0.35	4.7
MgSO_4_	0.30	1.2
CaCl_2_	0.17	1.2
KH_2_PO_4_	0.16	1.2
D-Glucose	1.50	8.3
Modified Krebs–Henseleit buffer used in Ussing chamber		
NaCl	6.72	115.0
NaHCO_3_	2.10	25.0
Na_2_HPO_4_.2H_2_O	0.42	2.4
KCl	0.37	5.0
CaCl_2_.2H_2_O	0.17	1.2
MgCl_2_	0.11	1.2
NaH_2_PO_4_.H_2_O	0.05	0.4
Mannitol	0.36	2.0
D-Glucose	1.80	10.0
HEPES	1.19	5.0
Kanamycin sulphate	0.10	0.2

^1^ Buffer pH 7.4.

**Table 2 nutrients-14-02524-t002:** Composition of short-chain fatty acids (SCFA) used in organ bath.

Item	Concentration (µmol/mL Buffer in Chamber)	Proportion (%)
Total SCFA ^1^	70.53	-
Acetate	50.36	71.40
Propionate	7.72	10.95
Isobutyrate	0.5	0.71
Butyrate	10.18	14.43
Isovalerate	0.52	0.74
Valerate	0.99	1.40
Caproate	0.26	0.37

^1^ SCFA = short-chain fatty acids.

**Table 3 nutrients-14-02524-t003:** Concentrations of short-chain fatty acids (SCFA; µmol/mL fluid) detected in fluid collected from the fetal stomach.

Item ^1^	Mean ^2^	SD ^3^
Total SCFA	0.35	0.18
Acetate	0.34	0.19
Propionate	0.01	0.02

^1^ Butyrate, isobutyrate, valerate, isovalerate and heptanoate were not detected. Caproate (0.01 µmol/mL) was detected in one sample. ^2^ Mean values of SCFA in the fluid collected from the stomachs of seven fetuses. ^3^ SD = standard deviation.

**Table 4 nutrients-14-02524-t004:** Basal electrophysiological measurements of the fetal porcine jejunum and changes in the tissue response to the addition of SCFA concentrations ^1,2^.

Item	Control	SCFA	SE	*p*-Value
Basal I_SC_ (µA/cm^2^)	−37.5	−32.4	6.68	0.395
Post-additional I_SC_ (µA/cm^2^)	−37.2	5.1	4.77	<0.001
Difference in I_SC_ (µA/cm^2^)	0.4	37.5	4.78	<0.001
Difference in I_SC_ (% of basal I_SC_)	1.0	126.8	4.77	<0.001
Basal G_T_ (mS/cm^2^)	20.0	19.9	1.50	0.947
Post-additional G_T_ (mS/cm^2^)	19.0	11.9	1.21	<0.001
Difference in G_T_ (mS/cm^2^)	−1.05	−8.02	0.53	<0.001
Difference in G_T_ (% of basal G_T_)	−5.1	−40.8	1.60	<0.001

^1^ SCFA = short-chain fatty acids; I_SC_ = short-circuit current; G_T_ = transepithelial conductance. ^2^ Values are least squares means and standard error of means (SE), where *n* = 16.

**Table 5 nutrients-14-02524-t005:** Gene copy numbers (log_10_ gene copies/25 ng RNA) of target genes in fetal jejunal tissue.

Gene of Interest	Mean	SE ^1^
*FFAR2*	1.6	0.02
*FFAR3*	2.1	0.02
*HCAR2*	0.2	0.08
*HDAC1*	3.8	0.01
*TLR2*	2.6	0.03
*NLRP3*	2.4	0.03
*EGFR*	3.6	0.02
*PPARG*	2.4	0.03
*SMCT1*	3.8	0.03
*SMCT2*	2.9	0.04
*MCT1*	4.0	0.02
*ZO1*	3.8	0.02
*OCLN*	3.9	0.03
*CLDN1*	1.8	0.05
*CLDN4*	4.9	0.04
*CDH1*	4.1	0.02
*JAML*	0.3	0.04
*NFKB*	0.9	0.17
*BD3*	2.3	0.02
*EP2C*	2.0	0.01
*IL10*	2.4	0.02
*IL18*	3.0	0.03

^1^ SE = standard error of the mean.

## Data Availability

The data presented in this study are available on request from the corresponding author.

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
