# Peer review of "Short-Chain Fatty Acids Modulate Permeability, Motility and Gene Expression in the Porcine Fetal Jejunum Ex Vivo"

_nutrients, 2022, doi:10.3390/nu14122524_

Round 1

Reviewer 1 Report

This study aimed to evaluate whether the effects of SCFA on muscle tension and epithelial permeability and related signaling in jejunal tissue from the porcine fetus in late gestation. It showed SCFA reduces mucosal pereability and muscle contraction. The work is well described and written and unveil the importance of CHO and amino acid derived metabolites for fetal gut maturation and development.

Introduction : please define what SCFA you are refering to. Please empahsize about their mode of actions as they can differ.

Table 4 : I think you can put it in supplementary material as it is quite large

Table 5 : Difference in ISC (% of basal ISC) and Difference in GT (% of basal GT) are difficult to understand due to negative values. I think you can remove

Figure 3, Figure 4 : colour legend of the histograms is missing

Discussion : SCFA used in the study is a mix, do you know which SCFA could drive the response ?

It is unfortunate that concentrations tested in the study are much higher than the ones found in amniotic fluid

Author Response

Comments and Suggestions for Authors

This study aimed to evaluate whether the effects of SCFA on muscle tension and epithelial permeability and related signaling in jejunal tissue from the porcine fetus in late gestation. It showed SCFA reduces mucosal pereability and muscle contraction. The work is well described and written and unveil the importance of CHO and amino acid derived metabolites for fetal gut maturation and development.

Authors: Thank you very much for your comments and suggestions.

Introduction : please define what SCFA you are refering to. Please empahsize about their mode of actions as they can differ.

Authors: We have specified what kind of SCFA we were referring to in the Introduction section (L50-52; L 54-60).

Table 4 : I think you can put it in supplementary material as it is quite large

Authors: Table 4 was moved to the supplementary material as Table S1.

Table 5 : Difference in ISC (% of basal ISC) and Difference in GT (% of basal GT) are difficult to understand due to negative values. I think you can remove.

Authors: Thank you for this comment. However, presenting the difference as proportional difference is helpful as it provides the dimension of the change. The negative sign for the ISC was removed. This should be then less confusing for the reader (Table 4).

Figure 3, Figure 4 : colour legend of the histograms is missing

Authors: Thank you. The missing legend was added.

Discussion : SCFA used in the study is a mix, do you know which SCFA could drive the response ?

It is unfortunate that concentrations tested in the study are much higher than the ones found in amniotic fluid

Authors: Thank you for this comment. The receptors have indeed different affinities for the various SCFA. Therefore, the receptors probably responded to more or less all SCFA that were in the mix. However, testing one SCFA at a time would have given a different picture (456-460.

Because we wanted to guarantee a response of the mucosa, we utilized the for the fecal mucosa very high concentration of SCFA. Also, this approach might be close to the situation when external supplements would be given after premature birth in humans or animals such as the pig.

Author Response

 In their manuscript “Short-chain fatty acids modulate permeability, motility and 2 gene expression in the porcine foetal jejunum ex vivo” Metzler-Zebeli et al. present data on the effect of SCFA on porcine foetal jejunum on functional and gene expression level. The manuscript is well written, the introduction gives sufficient background and nicely explains the rationale for the study. Tables and results appear to be written in some haste, as there are slips such as missing units and could have been prepared with more consideration. The discussion shows a critical understanding of the topic. The study is relevant and of interest to the field. Before the manuscript can be considered for publication, however, several points of concern should be clarified.

Authors: Thank you for your critical comments and suggestions to improve our manuscript.

Major concerns:

1) A major point of criticism is the small number of animals (N = 4) enrolled in the study. The authors try to compensate this by doing the statistical analysis based on the number of technical replicates (n = 16). Therefore, I strongly recommend to a) redo the statistical analysis based on the mean values of the technical replicates for each animal (N = 4) and b) consider whether there is a possibility to add at least two more animals to the study.

If I misunderstood and there were 4 piglets each per sow, i.e., a total of N = 16 animals, I apologize but then this should be made clear in the manuscript then, as I am only guessing this possibility based on well-meaning. Generally, the number of technical replicates per animal (e.g., for qPCR) are not stated clearly in the current form.

Authors: Thank you for your comment. There seems to be a misunderstanding as the presented work is not a “classical” in vivo trial. Just for correctness, that were not technical replicates but biological replicates (chamber replicates). This means that, regarding the experimental design of the present study, it is important to consider that this ex vivo experiment demanded a different experimental design compared to classical in vivo trials. The sample size is a factor of a certain number of different animals and several (gut tissue) replicates per animal. With regards to the experimental unit in our SAS model, this is why we used the term ‘chamber replicate’ to clarify the actual experimental unit. The number of observations in this type of ex vivo study is based on the number of biological replicates within an animal. The power analysis that we performed before this experiment showed that 16 replicates originating from 4 animals will bring sufficient power to reject the null hypothesis.

2) The authors explain the rationale for sampling the foetuses’ gastric fluid instead of original amniotic fluid. However, then it cannot be called amniotic fluid anymore, SCFA can not only be absorbed but even generated in the foetal gut instead of in the amniotic fluid, which could thus challenge a basic conclusion of this study. Please change the wording for that accordingly.

Authors: We changed our wording in the text to make it more obvious that we collected the fluid from the stomach. For explanation, it is almost impossible to collect amniotic fluid in a sterile manner directly during necropsies from the porcine uterine horns. Please consider that there are about 20 fetuses in a sow and the situation differs from that of humans where the amniotic fluid can be collected via a needle through the abdominal wall. Often the myometrium and the fetal membranes are already open and broken, respectively, and the amniotic fluid is not sterile anymore (L112-117).

3) The calculation of the gene expression seems to be somewhat unusual. If I understood correctly, the authors used one single sample as a reference for all the other samples. This appears to be random, especially considering that later in the manuscript they state to have measured “absolute” expression levels. Please reconsider using a mean value, e.g., of all control samples, as a reference. You should then be able to compare the expression levels within and also between animals.

Authors: There seem to be some misunderstandings about the calculation of the relative and absolute gene expression. The calculation of the relative and absolute expression levels were different. For the relative expression, we used the classical 2 –ddCT method which is a valid and widely used method to assess the relative gene expression among treatment groups. In applying this method, as we presented in detail in the Material and Methods section, there were two normalization steps: “the geometric mean of the HKG was used for normalization of the Cq values to determine ΔCq values. The sample with the highest expression and hence lowest ΔCq value of the respective targeted gene was used for the second normalization in order to calculate the ΔΔCq and respective 2-ΔΔCq.”

The absolute gene expression was calculated using serial dilutions of known concentrations of cDNA for each primer set: “For the absolute gene expression, standard curves were generated using serial dilutions (10-3 to 10-7 molecules/µL) of the purified products (QIAquick PCR Purification Kit, Qiagen, Hilde, Germany) and quantified PCR products (Qubit™ dsDNA HS Assay Kit, Thermo Fisher Scientific, Waltham, USA) generated by qPCR from the samples to as-sess amplification efficiency (Table S1).”

4) The authors state that a) the epithelium could not be isolated from the muscularis and serosa and b) that addition of SCFA decreased muscular tone of the muscularis. Hence, there might be an important influence of muscular tone on electrophysiology in the Ussing chamber experiments as well. This should be discussed.

Authors: There seems to be a misunderstanding about the two different methodological approaches that we used, the organ bath and Ussing chamber system. Commonly, there is no stripping of the serosal layers for the Organ bath system. The gut tube piece is used as it is. The serosal layers do not interfere in the measurements of the muscle tension.

 For Ussing chamber experiments it is advisable to remove the outer serosal layers as they caninterfere in the flow of current through the tissue. However, for the fetal tissue this was not nessesary as the serosal layers were very thin and the risk exist when removing the serosal layers that the mucosa breaks. For the Ussing chamber experiment a tissue piece of 0.91 cm2 was used that was fixed between the half chambers with tiny needles. From the experimental set-up, there is no effect of the muscular tone on the electrophysiological measurements.

Minor points:

- L. 50: I would suggest “maternal blood” instead of “mother’s blood”

Authors: Done (L 50).

- L. 74: similar deficiencies

Authors: Done (L 78).

- L. 86: the first sentence is unnecessary in this context

Authors: Done (L 90). The first sentence was removed from the M&M section.

- L. 100: the medication and dosage used for sacrificing the pigs should be given here. How was the death of the piglets assured?

Authors: This information was provided (L101-110)

- L. 134 ff. incl. table 3: in my opinion, this should rather be part of the results (table 3) and the discussion. Table 3 is lacking units for the data given.

Authors: The missing units were added. Table 3 was mentioned in the M&M section, but it was already a part of the Results and Discussion sections.

- L. 201ff.: the amplicons are rather large for a conventional qPCR, what is the reason for choosing these primer pairs? Were the primers designed to be intron-spanning or was contamination with genomic DNA excluded by other means? According to the MIQE-guidelines, the use of the term “reference genes” is to be preferred.

Authors: Thank you for this comment. The term housekeeping genes was replaced by reference genes. With regard to the amplicon length, for most genes short amplicons could be achieved. However, for some genes, it was difficult to find suitable primer sets for short amplicon length that worked properly. We commonly do evaluation runs for the primer sets, we are going to use in a qPCR experiments, and we do this for each new qPCR project. So the primer sets were evaluated to amplify properly. In terms of the intron-spanning, primer sets were designed to be intron-spanning, whenever the available sequences in gene banks allowed the proper design with intron-spanning.

- L. 272: the control chambers within 30 min of the test period?

Authors: Corrected (L284).

- Table 5: Isc values are certainly not negative?

Authors: The actual basal Isc values were negative in both groups (which is the normal situation for the porcine jejunum). However, when presenting the delta Isc values, we removed the minus sign.

- Table 6: how did you measure absolute expression values and which incubation groups were the samples from? In the M&M section, only RT-qPCR and ddCt is explained. What is the message of this table anyway?

Authors: There must have been a misunderstanding. The calculation of the absolute quantification was provided in the respective M&M section.

- Why did you choose OCLN, CLDN-4, CDH and JAML as tight junction proteins worth to be investigated? What about ZO-1, other claudins, esp. pore forming claudins, etc.? Please add an explanation to the discussion.

Authors: Thank you for this comment. We investigated ZO-1 expression. Please see Table 5 and Figure 4. The selection of tight-junction proteins was based on results from previous studies from our groups.

- L. 441ff.: it seems very bold to judge which signalling pathways are being activated by SCFA from assessing the gene expression of receptor proteins only. Also, the conclusion, that “functional protein levels” (l. 473f.) are affected cannot be drawn from the results of the study. To prove the recruitment or activation of transport protein, inhibitor studies would have been necessary. This should be phrased more carefully to avoid wrong deductions from the study.

Authors: Thank you for this comment. As we pointed out in the Discussion section, there are multiple signaling routes. However, we did not only investigate the receptor expression on transcript level but also the effect of SCFA on the functioning tissue so that we can deduce that a certain signaling occurred. Also, we mentioned in our discussion that we very likely had negative feedback loops at 30 min of incubation of the tissue with a high SCFA concentration. We added a sentence to the conclusion to point out that inhibitor studies should be conducted (L496-497).

- L. 444: have been shown?

Authors. Done (L456).

Round 2

Reviewer 2 Report

Authors have addressed all questions of the reviewer.

As a personal remark I would like to question the practice of "waiting to ensure the death of each fetus". Although this was obviously approved by the local authorities it would be more ethical to actively kill each fetus instead of waiting for it to suffocate.